# Human Copper-Containing Amine Oxidases in Drug Design and Development

**DOI:** 10.3390/molecules25061293

**Published:** 2020-03-12

**Authors:** Serhii Vakal, Sirpa Jalkanen, Käthe M. Dahlström, Tiina A. Salminen

**Affiliations:** 1Structural Bioinformatics Laboratory, Biochemistry, Faculty of Science and Engineering, Åbo Akademi University, Tykistökatu 6A, FI-20520 Turku, Finland; serhii.vakal@abo.fi (S.V.); Kathe.Dahlstrom@abo.fi (K.M.D.); 2MediCity Research Laboratory, University of Turku, Tykistökatu 6A, FI-20520 Turku, Finland; sirpa.jalkanen@utu.fi

**Keywords:** copper-containing amine oxidases, vascular adhesion protein-1, diamine oxidase, inhibitor design, protein-inhibitor interactions, computer-aided drug design

## Abstract

Two members of the copper-containing amine oxidase family are physiologically important proteins: (1) Diamine oxidase (hDAO; AOC1) with a preference for diamines is involved in degradation of histamine and (2) Vascular adhesion protein-1 (hVAP-1; AOC3) with a preference for monoamines is a multifunctional cell-surface receptor and an enzyme. hVAP-1-targeted inhibitors are designed to treat inflammatory diseases and cancer, whereas the off-target binding of the designed inhibitors to hDAO might result in adverse drug reactions. The X-ray structures for both human enzymes are solved and provide the basis for computer-aided inhibitor design, which has been reported by several research groups. Although the putative off-target effect of hDAO is less studied, computational methods could be easily utilized to avoid the binding of VAP-1-targeted inhibitors to hDAO. The choice of the model organism for preclinical testing of hVAP-1 inhibitors is not either trivial due to species-specific binding properties of designed inhibitors and different repertoire of copper-containing amine oxidase family members in mammalian species. Thus, the facts that should be considered in hVAP-1-targeted inhibitor design are discussed in light of the applied structural bioinformatics and structural biology approaches.

## 1. Introduction

In humans, the copper-containing amine oxidase (CAO) family consists of three proteins [1]. Diamine oxidase (hDAO; AOC1) prefers diamines and is involved in the degradation of histamine [2], while retina-specific amine oxidase (hRAO; AOC2) is a monoamine oxidase with enzymatic activity in retina [3], and vascular adhesion protein-1 (hVAP-1; AOC3) is a multifunctional cell-surface receptor and an enzyme with preference for monoamines [4]. Both hDAO and hVAP-1 are physiologically essential proteins, but little is known about the physiological function of hRAO. In the CAO-catalyzed reaction, primary amines are converted to the corresponding aldehydes accompanied by the production of hydrogen peroxide and ammonia (for a review, see [5]). The structural characterization of human CAOs has revealed a dimeric protein with a deeply buried active site characterized by highly conserved residues and several common features involved in the catalytic reaction, including a topaquinone (TPQ) cofactor and a conserved aspartate residue, as well as three conserved histidine residues that bind the copper ion crucial for the TPQ biogenesis (hVAP-1: [6,7,8,9,10]; hDAO: [11,12,13] (Figure 1A,B). Three arms that protrude from one monomer to the other one stabilize the tight dimer of human CAOs, where each monomer consists of three domains, namely, D2, D3, and D4 (Figure 1A). The D4 forms the deeply buried catalytic site, and Arm I of monomer B (pink) contributes to ligand binding into monomer A (cyan) (Figure 1B).

In addition to its enzymatic activity, hVAP-1 is an adhesion protein involved in leukocyte trafficking, whose roles in leukocyte adhesion and inflammatory conditions are well characterized (for review see [14,15,16,17]). In adhesion, Sialic acid-binding immunoglobulin-like lectin 9 (Siglec-9), a leukocyte surface receptor [18], forms multivalent interactions with hVAP-1 on the endothelial cell surface [19]. Typically, healthy humans have a low level of hVAP-1 activity, but the levels are elevated upon inflammation when hVAP-1 is translocated onto the luminal surface of the endothelium [20,21,22]. Therefore, molecules that bind to hVAP-1 can be used as tracers in positron emission tomography (PET) to detect the site of inflammation in vivo [18]. Additionally, the anti-inflammatory potential of VAP-1 inhibitors is shown in several inflammation models (reviewed in [17]), which have proven that the enzymatic activity of hVAP-1 is crucial for leukocyte adhesion and especially required for the hVAP-1-dependent extravasation in vivo [23]. Moreover, anti-hVAP-1 antibodies, which do not affect the enzymatic activity of hVAP-1, independently inhibit hVAP-1-mediated adhesion. Thus, both antibodies and small molecular inhibitors could be used as anti-inflammatory drugs to treat acute and chronic inflammatory conditions. There is also substantial evidence that hVAP-1 inhibitors could be utilized in preventing tumor progression and metastatic spread of cancer (reviewed in [14,17]). 

hDAO is the main enzyme in the extracellular degradation of histamine [24]. It is expressed in the kidney, placenta, intestine, thymus, and seminal vesicles [25], and its functions are related to cell proliferation, inflammation, allergic response, and ischemia [11]. Decreased hDAO levels have been linked to histamine intolerance [26], and a decreased hDAO activity coupled to a high presence of histamine increases the probability for anaphylactic shock-like symptoms [27]. Normally, the plasma concentration of hDAO is elevated during pregnancy, and low levels of hDAO increase the risk of premature termination [28]. It is known that some pharmaceuticals, which are not targeted towards hDAO, inhibit its normal function (off-target effect) and thus cause harmful side effects (adverse drug reactions; ADRs) [11,27]. Consequently, it is a crucial drug safety issue to identify substances, which potently inhibit hDAO activity, to avoid their usage in the treatment of patients with hDAO deficiencies and conditions like varieties of urticaria, asthma, atopic dermatitis, mastocytosis, mastcell activation syndromes, and likely, pregnant women.

This review concentrates on the structural chemistry of hVAP-1 inhibitors whose binding site has been either predicted by computational studies or deciphered by X-ray crystallography. The current knowledge of the off-target effect of hDAO inhibition is also summarized and discussed in light of the hVAP-1-targeted inhibitor design. Furthermore, we highlight important facts and considerations that need to be taken into account in the hVAP-1 inhibitor design process to obtain potent and selective drug molecules.

## 2. Catalytic Mechanism of Copper Amine Oxidases

VAP-1 is a primary amine oxidase and, thus, catalyzes the oxidative deamination of primary amines, both exogenous (allylamine, benzylamine) and endogenous (methylamine and aminoacetone). According to modern concepts [29,30,31], the reaction proceeds through a ping-pong mechanism. In the reductive half-reaction, TPQ is reduced by the substrate to generate the aldehyde product (Figure 2A); then, in the oxidative half-reaction, the reduced TPQ is reoxidized by molecular oxygen, which is accompanied by a release of hydrogen peroxide and ammonia [4,32] (Figure 2B). The overall reaction balance can be written as R-CH_2_-NH_3_^+^ + H_2_O + O_2_ → R-COH + H_2_O_2_ + NH_4_^+^.

The reductive half-reaction comprises three steps (Figure 2A); first, the oxidized TPQ reacts with a protonated primary amine to form a so-called “substrate Schiff base” (quinone imine), and then, this adduct is converted to a “product Schiff base” (quinolaldimine), which is facilitated by a conserved Asp-386 catalyzing proton abstraction [33]. In the final step, the aldehyde product (R-COH) is hydrolytically released generating the reduced TPQ (aminoquinol).

TPQ coordinates the Cu(II) ions; two states named “off”-copper and “on”-copper have been characterized and shown to exist in equilibrium in solution [34]. In the “off”-copper conformation, oxidized TPQ does not interact with Cu(II), which is coordinated by three histidine residues and two water molecules, while in the “on”-copper state, the waters are displaced, and TPQ interacts with the copper.

The oxidative half-reaction proceeds through four steps via one of two possible ways (Figure 2B). In both scenarios, the intermediate molecules are the same, but the electron transfer pathway differs. Currently, it is not clear whether molecular oxygen reacts directly with reduced TPQ, TPQ semiquinone, or Cu(I) [31] as there is evidence for each option. In the first scenario, oxygen interacts with Cu(I) bound to TPQ-semiquinone and then is reduced to a reactive O_2_^•–^ via inner-sphere electron transfer (Figure 2B, upper scheme) [35]. Alternatively, according to Mure et al. [36], oxygen binds to hydrophobic residues near the reduced TPQ, which reduces O_2_ to O_2_^•−^ via outer-sphere electron transfer. Then, in both scenarios, Cu(II)-hydroperoxide and TPQ-iminoquinone are generated [37]. At the final step, iminoquinone is hydrolyzed, and ammonia is released with the concomitant production of hydrogen peroxide and oxidized TPQ-Cu(II) regeneration.

Recent kinetic and computational studies suggest that a redox-active metal in copper amine oxidases is needed to catalyze the reduction of O_2_ to H_2_O_2_ [38]. Cu(I) reacts with O_2_ by inner-sphere electron transfer to eliminate charge accumulation due to O_2_^•−^, thus lowering the free energy barrier by minimizing the outer-sphere reorganization energy [39]. Copper also facilitates long-range proton-coupled electron transfer to bind oxygen via the H-bond network [38]. However, it should be noted that most of the data on catalytic details have been acquired using non-mammalian bacterial, plant, and fungal amine oxidases, primarily *Escherichia coli*, *Arthrobacter globiformis*, and *Hansenula polymorpha* orthologs.

## 3. Medical Relevance of Targeting hVAP-1 

### 3.1. Basis for Clinical Targeting of VAP-1

Due to the multifunctional nature of VAP-1 and its involvement in inflammation via adhesive leukocyte–endothelial cell interactions and production of end-products, which modulate other adhesion and signaling molecules triggering inflammation, VAP-1 inhibition provides us with a novel approach to overcome several diseases having inflammatory components. Moreover, the enzymatic activity of VAP-1 modifies its substrate to an aldehyde that is able to promote the formation of advanced glycation end-products damaging vasculature, for example, in diabetes. Indeed, a multitude of preclinical studies using mice, rats, and rabbits have shown beneficial effects of targeting VAP-1 in several disease models. They include autoimmune and other inflammations, viral and bacterial infections, ischemia-reperfusion injuries, fibrosis, cancer, and metabolic diseases (reviewed in [17]). These studies, together with the findings that hVAP-1 is translocated to the endothelial cell surface from intracellular storage granules at sites of inflammation and that increased concentrations of hVAP-1 are found in several diseases, form the basis for clinical targeting of VAP-1 (reviewed in [17]). Moreover, easy accessibility of VAP-1 on inflamed endothelium for potential imaging agents makes VAP-1 an optimal target to search for inflammatory foci that is often challenging in clinics.

### 3.2. Clinical Trials

After the discovery of leukocyte ligands for VAP-1 [18,40], a VAP-1 binding peptide of Siglec-9 has been developed as a novel imaging agent. This peptide binds to VAP-1-positive vessels in rheumatoid synovium [41], and the peptide conjugated with ^68^Ga-Dota has just recently successfully passed the phase I clinical trial [42]. It is going to further clinical trials meant to test this peptide as a diagnostic and follow-up tool for arthritic lesions in PET imaging.

Several companies are also developing therapeutics to block the function of hVAP-1. They include both antibodies and small molecular inhibitors (Table 1). Currently, active clinical trials or completed trials with accessible information about their outcome are discussed below. BioTie Therapies (currently Acorda) developed a fully human anti-hVAP-1 antibody Timolumab (BTT1023). It has been well-tolerated and shown efficacy both in early clinical trials for rheumatoid arthritis and psoriasis. In contrast, a phase 2 proof-of-concept trial for 19 patients suffering from primary sclerosing cholangitis did not meet the pre-defined efficacy criteria in the interim analysis, and the trial was terminated [43].

Astellas’ ASP8232 VAP-1 inhibitor was tested in placebo-controlled phase II trials for diabetic macular edema and diabetic nephropathy [44,45]. In macular edema, this inhibitor was not effective, although it efficiently inhibited VAP-1 enzymatic activity in plasma. How well the orally administered inhibitor reached the vitreous fluid was not tested and the authors speculate that the route of delivery should be reassessed in the future trials. In contrast, 12-week daily treatment with orally administered ASP8232 significantly reduced albuminuria of diabetic patients. The results of this trial suggest that diabetic kidney disease can be delayed by inhibiting VAP-1 activity. In both of these trials, ASP8232 was safe and well-tolerated. 

Pharmaxis developed BXS4728A to inhibit VAP-1 activity and performed a phase I clinical trial. Moreover, this compound turned out to be safe and well-tolerated. It is orally bioavailable and has long-lasting inhibitory effects. Boehringer-Ingelheim acquired this compound and tested it in a phase II trial for non-alcoholic steatohepatitis (NASH) that is a growing health problem connected to obesity and type 2 diabetes and can result in permanent liver damage. The study met its pre-specified endpoints including a decrease in NASH biomarkers but due to potentially harmful interactions with a drug (not defined in the press release) used by NASH patients, the company will not continue to develop the inhibitor to this group of patients. However, they continue to run phase II trials for diabetic retinopathy, but in this multi-center double-blind study involving 100 patients with moderately severe non-proliferative diabetic retinopathy, the diabetic macular edema is not targeted [46]. Currently, also Terns Pharmaceuticals runs phase I trial for NASH with its VAP-1 inhibitor, TERN-201 (https://www.ternspharma.com/pipeline). Besides these, several other companies have VAP-1-targeted programs in their pipeline for pulmonary diseases, pain, uveitis, multiple sclerosis, inflammatory bowel diseases, Alzheimer’s disease, and Parkinson’s disease at preclinical or inactive state. Which one of these will end up in clinical trials remains to be seen. 

## 4. Structural Biology and Computer-Aided Drug Design in hVAP-1-Targeted Inhibitor Design 

As hVAP-1 is a suitable drug design target, several potential hVAP-1 inhibitors have been published including hydrazines, thiazoles or other nitrogen-containing heterocyclic cores, allylamines and propargylamines, amino acid derivatives, benzamides, oxime-based inhibitors, aminoglycoside antibiotics, vitamin B1 derivatives, and peptides (Reviewed, e.g., in [32,75,76]). Nevertheless, most of the current hVAP-1 inhibitors are mechanism-based and bind irreversibly to the TPQ cofactor in a similar way as 2-hydrazinopyridine (2HP; PDB code 2C11) [7] (Figure 3). The irreversible binding mode is an undesirable feature for human drugs, since only new protein synthesis restores the function [77]. Thus, the design of reversible inhibitors specific for hVAP-1 would be highly beneficial. So far, pyridazinones (Figure 3; PDB codes 4BTY, 4BTX, and 4BTW; [10]), to our knowledge, are the only published inhibitors designed to have a reversible binding mode to hVAP-1. Figure 3 simultaneously shows both mechanism-based and reversible binding sites and modes via the overlaid 2HP and pyridazinone complex structures. It serves as a binding-site map for further discussion on the binding pose of the reviewed inhibitors for which the structural data is not publicly available (Table 1 and Table 2). Computer-aided drug design has been reported for several mechanism-based inhibitors (Table 2), and the results have provided foundations for some of the inhibitors that have entered clinical trials and/or have been extensively used in preclinical studies (Table 1). 

### 4.1. X-Ray Structures of hVAP-1 Complexes for Docking Studies and Inspiration for Drug Design

Most of the docking studies (Table 2, Appendix A) have been conducted using the 2-hydrazinopyridine (2HP) complex of hVAP-1 as the target protein (PDB code 2C11) [7] and, thus, the focus is on the special features of this structure and its comparison to other solved structures. Moreover, the crystal structures of hVAP-1 in complex with imidazole (PDB codes 2Y73 and 2Y74; [9]) are presented as they unintentionally have provided valuable data on the inhibitor binding properties of hVAP-1. All publicly available X-ray structures of hVAP-1 and hDAO are listed in Table 3.

#### 4.1.1. Irreversible Complex of 2HP with hVAP-1

The X-ray structure for the extracellular part (residues 29-763) of hVAP-1 in complex with 2HP was solved at 2.9 Å resolution by Jakobsson et al. in 2005 (PDB code 2C11; [7]). The crystals were obtained by soaking the crystals of the holoenzyme (PDB code 2C10) with 5 mM CuCl_2_ and 8 mM 2HP. Due to the addition of CuCl_2_, additional Cu^2+^ ions were detected, and one of them interacts with the residues in the Arg726-Gly725-Asp728 hairpin loop, changing its conformation, which likely causes the lack of 34 C-terminal residues (729-763). The resulting complex structure also lacks large portions of the N-terminal residues 29-57. Despite these non-natural features, the 2HP adduct in the complex structure exists primarily as a hydrazone and shows how 2HP interacts with the catalytic site (Figure 4A). The TPQ is in off-copper conformation, where 2HP may react with C5 of TPQ. The catalytic Asp386 forms hydrogen bonds with the N2 and N3 nitrogens of 2HP, which stacks with Tyr384 and Phe389. Furthermore, Leu468 and Leu469 form hydrophobic interactions with 2HP. Since the computationally designed ligands have been planned to interact covalently with TPQ, the 2HP adduct structure has been most frequently used as a target in docking studies (Table 2).

#### 4.1.2. Diverse Binding Modes Between Imidazole and hVAP-1 

Two structures in complex with imidazoles were obtained when soluble hVAP-1 was extracted from human serum and crystallized in imidazole containing buffer [9]. Both of them similarly interact with two imidazoles that inhibit the enzymatic activity; the first one by binding to TPQ and the second one by blocking access to the active site. Both structures were crystallized under similar conditions but, interestingly, have the TPQ cofactor in different conformations. In one of them, TPQ is in the off-copper conformation, and the imidazole is covalently bound to TPQ (Figure 4B; PDB code 2Y74), whereas in the other one TPQ is in the on-copper conformation and the imidazole binds non-covalently to it (Figure 4C; PDB code 2Y73). Compared to the 2HP complex (Figure 4A), the first imidazole makes only one hydrogen bond with Asp386 and stacks between Tyr384 and Leu468 in both complexes (Figure 4B,C). Despite the different conformation of TPQ, the binding site of the non-covalently bound imidazole totally overlaps with the covalently bound imidazole and both N1 and N3 form hydrogen bonds to O_2_ of TPQ and Asp386, respectively (Figure 4C). The second imidazole (yellow; Figure 4D) stacks with Tyr176, makes hydrophobic interactions with Leu469, Phe368, and Leu447 of Arm I from the other monomer, and its N3 forms a hydrogen bond with the hydroxyl group of Tyr394. In both the off-copper (Figure 4D) and on-copper (Figure 4E,F) structures, the second imidazole has the same binding site but the hydrogen bonding network with Thr212 is different in monomer B of the on-copper complex (Figure 4F). In monomer A of both complexes and monomer B of the off-copper complex, the N1 nitrogen makes a water-mediated hydrogen bond with the main chain nitrogen of Thr212 (Figure 4D,E). In chain B of the on-copper complex, the side chain of Thr212 has a different conformation, and its hydroxyl group forms an additional hydrogen bond with the water molecule that interacts with N1 of the second imidazole (magenta, Figure 4F). Thus, the binding mode of the second imidazole differs in the monomers of hVAP-1 when the first imidazole is bound non-covalently to on-copper TPQ (Figure 4D,F). The closest distance between the two imidazoles is 7.5 Å.

### 4.2. Mechanism-Based Inhibitors 

#### 4.2.1. Hydrazines 

Nurminen et al. (2011) [78] used the 2HP–hVAP-1 complex as a target for covalent docking of modified hydrazines (Table 3). The stacking interactions with Tyr384 and Phe389 formed by compound **R1** are equal to those formed between 2HP and hVAP-1 (Figure 4A), but the additional hydroxyl group of **R1** forms a hydrogen bond with the catalytic Asp386. The addition of a larger hydrophobic group to the C2 position of the hydrazines increased the selectivity of the inhibitors towards hVAP-1 over monoamine oxidase (MAO). Compounds **R2** and **R3** differ by the N-methyl group of **R3**, which changed the binding mode in such a way that the methyl group points towards Leu469. Therefore, the hydrogen bond between the same nitrogen and Asp386 that occurs in **R2** is lost in **R3**, and this allows better hydrophobic interactions for the C2 substituent in **R3**. Thus, computational modeling explained the better binding affinity of **R3** compared to **R2**. The docking studies combined with molecular dynamics simulations revealed that the flexibility of Met211, which is part of the hydrophobic pocket formed by Tyr384, Phe389, Tyr394, Leu468, and Leu469 (Figure 4A), allows accommodation of larger hydrophobic groups and even aromatic ring (compounds 11a–d) to the C2 position. La Jolla Pharmaceutical has been developing another hydrazine compound, namely LJP1207, with phenylpropenyl moiety attached to the hydrazine group. Initially, it showed promising results in a series of animal models, including autoimmune encephalomyelitis [52], chronic inflammation modeling ulcerative colitis [53], transient forebrain ischemia [86], and liver cancer [87]. However, due to its potentially toxic allylhydrazine structure, further development of LJP1207 was discontinued. In general, the safety issues related to the development of hydrazine derivatives have inspired hVAP-1 targeted inhibitor development towards other types of inhibitors [32].

#### 4.2.2. Thiazoles

Inoue et al. [79] identified a thiazole hit compound with an IC50 value of 3.5 μM for hVAP-1 inhibition using high-throughput screening (HTS) and structure-activity relationship (SAR) studies. The developed thiazole compounds have a guanidine moiety that mimics a substrate Schiff base intermediate complex with TPQ. In accordance, the best compound **R5** of the study had unsubstituted quinidine and 0.23 μM hVAP-1 inhibition. When **R5** was covalently docked to the N1 nitrogen of the 2HP adduct (PDB code 2C11), the guanidine moiety established a tight hydrogen-bonding network with Asp386 providing a structural explanation for its importance as an unsubstituted functional group (Table 3). The sulfur atom in the thiazole ring formed a hydrogen bond with the backbone nitrogen of Thr210 and the amido group a proton–π interaction with Tyr176. Compound **R5** was further analyzed by an in vivo test, and it was proved to be an inhibitor of macular edema [81]. It also exhibited improved binding to rat VAP-1 (rVAP-1) and more than 435-fold selectivity towards VAP-1 over DAO and MAO-B. The fact that compound **R5** had significantly lower VAP-1 inhibitory activity in humans than in rats encouraged Inoue et al. [80] to develop the compound further. They used molecular docking together with pharmacophore modeling and combined features of **R5** and another compound from HTS, which did not have a guanidinyl group. Series of thiazole and pyrazole compounds were subjected to SAR studies where thiazole was found to have a better VAP-1 inhibitory potency over pyrazole. The results were also analyzed in light of the sequence comparison of rat and human VAP-1. As a result, compound **R6** was first identified but it established weak inhibitory activity for rVAP-1. Based on the sequence analysis and docking results, the methanesulfonylphenyl moiety of **R6** locates near Leu447, and the corresponding Phe447 in the rVAP-1 causes a steric hindrance for binding. Thus, they designed compound **R7** (U-V002) by changing the phenyl group in **R6** to benzyl group, which increased inhibitory activity for rVAP-1 50-fold and hVAP-1 about 2-fold [80]. U-V002, developed by R Tech Ueno Ltd.^®^ showed promising inhibiting potency towards both human and rat VAP-1 with IC50 ≈ 7 nM and 8 nM, respectively, with low effect on MAO-A and MAO-B (IC50 > 10 µM for both) [70]. U-V002 also showed promising anti-inflammatory and anti-neovascularization effects in animal models [71,73] and, furthermore, it was evaluated preclinically for the treatment of diabetic retinopathy [72]; unfortunately, current development status of the compound is unknown.

#### 4.2.3. Indanols

Biotie Therapies Corporation developed a series of hydrazine indanols of which BTT2052 was a potent inhibitor of hVAP-1, and its binding mode was computationally studied by manual docking to TPQ [74]. It binds like other hydrazines (Figure 3A) and, similarly to the hydroxyl group of the C2 substituted hydrazines (4.2.1), the hydroxyl group of indanol forms a hydrogen bond with Asp386, while the rings form hydrophobic interactions with Met211, Tyr384, Phe389, and Leu469. Despite BTT2052 being a potent inhibitor of hVAP-1 and not inhibiting MAO, it was an effective inhibitor of hDAO and hRAO [74]. This is understandable in light of our current knowledge of human CAO active site channels [1], showing that the residues in the catalytic site are highly conserved and residues in the active site channel contribute to the subtype specificity. As a small inhibitor, BTT2052 does not interact with these residues in the active site channel. The same considerations are valid in the case of a highly similar molecular entity, BTT-2027, which was shown to have high selectivity to hVAP-1 over twenty test targets with Ki ≈ 54 nM [20]. Although no modeling studies have been published, it can be hypothesized to bind tightly to hDAO and hRAO in the same orientation as BTT2052 does.

#### 4.2.4. *H*-Imidazol-2-Amines

Inoue et al. [81] continued their earlier work with thiazole compounds (4.2.2) intending to design compounds that bind better to hVAP-1, as the developed compound **R5** [79] was a much better inhibitor of rat than hVAP-1. They first used the thiazole pharmacophore of compound **R5** (named **2** in [81]) and optimized the phenylguanidine biostere, 1H-benzimidazol-2-amine (compound **R8**) to design compound **R9** by synthesis and SAR of 1H-imidazol-2-amines compounds. The inhibitor VAP-1 activity of **R9** was 19 nM towards human and 5.1 nM towards rat enzyme [81]. In their study, docking analysis of compound **R8** suggested that its benzene ring is not optimally positioned in the active site of hVAP-1. Based on these results, they added a thiazole substituent in the 4-position of the benzene ring to create compound **R9**. The docking studies of **R9** indicated that the stronger interactions formed by the aminothiazole moiety are responsible for its increased inhibitory activity compare to compound **R5** (Table 2). Compound **R9** was highly selective towards VAP-1 over other CAOs and MAOs, and it was orally active in a pathological rat model of macular edema [81]. Later, **R9** proved not to be suitable for further clinical development due to its insufficient pharmacokinetic properties in rats [83]. 

#### 4.2.5. Allylamines

Pharmaxis Ltd.^®^ published their first paper on allylamine inhibitors of hVAP-1 in 2012 [54]. Their project was inspired by the irreversible haloallylamine inhibitor of MAO-B, mofegiline, the inhibition mode of which is hypothesized to rely on a covalently attached reactive intermediate that is alkylated to release fluorine, thus leading to irreversible inhibition [54]. In their study, modeling and SAR were systematically used to design a highly potent compound **R10** (PXS-4159) with promising pharmacokinetic profile and good in vitro and in vivo inhibitory activity. It showed a high preference towards hVAP-1 as compared to MAO-A, MAO-B, and hDAO, which is evident from the IC50 values of 10 nM vs. >30, 1.64, and >30 µM, respectively [54]. Modeling showed a crucial role for the hydrophobic interactions between the cyclohexyl moiety and Phe173-Tyr394 for the stabilization of **R10** in the binding pocket. To the best of our knowledge, PXS-4159 is still subjected to the preclinical tests. After that, Pharmaxis reported several other haloallyamines (Table 1). In PXS-4728A (later renamed to BI 1467335), the cyclohexyl group was changed to the other lipophilic (isobutyl) moiety. In 2019, BI 1,467,335 completed the Phase II clinical trials for the treatment of non-alcoholic fatty liver disease; however, no results have been released yet. Another haloallylamine, PXS-4681A, which is no longer under development, has a highly polar sulfonamide group instead of the lipophilic cyclohexyl moiety, which can establish multiple hydrogen bonds with the same Tyr394 residue. Moreover, La Jolla Pharmaceutical Company^®^ has been evaluating haloallylamine LJP1586, which has high potency against both human and rat VAP-1 with IC50 values between 4 and 43 nM [22]. It showed promising results in animal models of subarachnoid hemorrhage-associated cerebrovascular dilating dysfunction [63], intracerebral hemorrhagic stroke [61], multiple sclerosis [60], and temporary middle cerebral artery occlusion [64]. However, no data on the further clinical development of the compound are available.

#### 4.2.6. Glycine Amides

Yamaki et al. (2017) [83] reported the development of their first VAP-1-targeted glycine amide derivatives as their previous work on thiazole substituted 1H-imidazol-2-amine (compound **37b** in Table 2) was not suitable for further clinical development. The first glycine amide scaffold with micromolar hVAP-1 inhibitor activity was identified by screening their in-house compound library, and its structural optimization was conducted using an extensive synthesis of glycine amide derivatives combined with SAR. Thereafter, they found that compound **R12**, a tertiary amide derivative of the initial compound, had increased stability but much weaker in vitro potency to human and rat VAP-1 [83]. At this point, molecular docking of **R12** with hVAP-1 was used to model the Schiff base intermediate with TPQ. Based on the molecular docking results, substituents to the 4-position of the most solvent-exposed phenyl ring were designed and synthesized. Based on the parallel SAR and artificial membrane permeability assay, compound **R13** (Table 2) showed the best properties and was selected for further studies. Next, the binding mode of **R13** was predicted using docking analysis and its binding pose shown to be similar to that of **R12**. The morpholine group at the 4-position of the phenyl ring protrudes from the active site cavity to the solvent between Arm I from the other monomer and Phe173 of D3 and the stacking interaction of the phenyl ring with Leu447 of Arm I is preserved like in **R12** [83]. Further studies of compound **R13** (Compound 1 in [84]) revealed that it had species-specific binding properties and was a better inhibitor of rat than human VAP-1. Similarly to the thiazole design, the sequence comparison of human and rat VAP-1 suggested that Leu447/Phe and Phe173/Thr differences are responsible for the weaker binding to hVAP-1. As the 4-position of the phenyl ring had been proven optimal in the previous study [83], replacement of the phenyl ring by various heteroaromatic rings was tested [84]. Several rounds of synthesis and SAR revealed compound **R14** with a nanomolar binding activity to human and rat VAP-1 and reduced CYP inhibition. The docking study predicted good stacking interactions between the first phenyl ring and Leu469, pyrimidine and Leu447 of ArmI, and piperazine and Tyr394. Furthermore, the chlorine of the surface phenyl group interacts with Asp446. Ex vivo and in vivo studies suggest that **R14** could be suitable for treating diabetic nephropathy [84].

### 4.3. Reversible Inhibitors 

#### 4.3.1. Pyridazinones 

The hVAP-1–imidazole complex structures (PDB core 2Y73 and 2Y74) revealed a novel, secondary binding site for imidazole in the active site channel, which inspired the design of reversibly binding pyridazinone inhibitors [10]. Three of the designed inhibitors were studied in more detail, and the X-ray structures in complex with hVAP-1 were solved for **R15** (IC50 70 µM; PDB code 4BTX), **R16** (IC50 290 µM; PDB code 4BTW), and **R17** (IC50 20 µM; PDB code 4BTY) (Table 3). The binding mode of the pyridazinones is similar and the 2.8 Å X-ray structure of the **R15**–hVAP-1 complex is shown as an example of the binding mode (Figure 5). For all the pyridazinones, the pyridazine ring stacks between Tyr176 and Leu447 from Arm I of the other monomer and makes hydrophobic interactions with Phe227. In addition, the carbonyl group of pyridazinone forms a hydrogen bond with the hydroxyl group of Tyr394 (Figure 5A and Figure 4B). The phenyl group in the inhibitors stacks with the aromatic rings of Phe389 and Tyr394 and forms hydrophobic interactions with Leu468 and Leu469 (Figure 5A). The hydroxyl group of Tyr448 originating from Arm I of the other monomer, the carbonyl group of Asp180, and the hydroxyl group of Thr210 form hydrogen-bonding networks that interact with the triazole nitrogen of the inhibitors. In the A chains, the methyl group of Thr212 makes hydrophobic interactions with the phenyl ring (Figure 5A), whereas, in all the B chains, the hydroxyl group of Thr212 forms a water-mediated hydrogen bond with the triazole nitrogen (Figure 5B). The variable part of the pyridazinones binds near the opening of the active site and forms hydrophobic interactions with Leu177 and Phe173 (Figure 5B). The binding sites of pyridazinones and the second imidazole overlap, and also, the binding mode in chain B of the on-copper TPQ complex (PDB code 2Y73) shares some similarities: Tyr394 and Thr212 are involved in the hydrogen bonding, and Tyr176 π-stacks with them (Figure 5C). Upon pyridazinone binding, the side chain of Phe173 changes conformation to accommodate the bulky ligand, and Leu177 moves slightly (Figure 5C). The TPQ, however, is in a different conformation in the non-covalent complexes of pyridazinone (off-copper TPQ) and imidazinone (on copper TPQ) (Figure 5C). 

A modeling study by Bligt-Lindén et al. [88] revealed the differences in the secondary imidazole binding site organization in human and monkey VAP-1 vs. rat and murine VAP-1. The width of the entry channel in rat and murine VAP-1 was limited by Phe447, which is not the case for the human and monkey enzyme with a smaller leucine residue. Besides, rodents have larger residues than primates in positions 177 (Gln vs. Leu) and 180 (Gln vs. Asp), which makes the channel even narrower. There are also additional differences in the composition of residues lining the active site channel that might affect ligand binding. Phe173 and Leu177 in hVAP-1 are changed to polar residues in rodent enzymes: Thr173 in mVAP-1, Asp173 in rVAP-1, and Gln177 in both rodent VAP-1s. Rats also have larger Thr in position 761 as compared to Ser761 in hVAP-1 and Ala761 in mVAP-1. All these residues are located at the entrance of the active site channel and might also affect ligand recognition. This finding can explain the high specificity of compounds **R15**, **R16**, and **R17** to hVAP-1 as compared to the murine VAP-1 (mVAP-1) [10]. For example, methylpiperazine moiety of **R17** is in close contact with residues 761 and 177, its phenyl ring interacts with residues 173, 177 and 447 and its imidazole ring with residue 180, which differs between human and rodent VAP-1. The largest and most lipophilic compound **R17** shows the best binding profile, which is in agreement with the hypothesis that primate VAP-1 prefers bulkier and more hydrophobic ligands than rodent VAP-1 [88].

#### 4.3.2. Phenyl-Piperidinyl Amine

Before the publication of pyridazinone complexes, Emanuela et al. (2012) reported docking and inhibitor activity studies of **R11** as a plausible skeleton for the design of reversibly binding inhibitors of hVAP-1 [82] (Table 2). Based on the docking results, **R11** (about 100 μM inhibitor) was predicted to have two different binding modes. In one of them, the inhibitor aligns with the opening of the active site cavity, reaching towards Asp180. In the other binding mode, one end of **R11** similarly aligns with the opening, but the other one intrudes to the active site and stacks between Phe389 and Leu469. 

## 5. hDAO as an Off-Target 

### 5.1. Biological Function 

DAO, with distinct specificity for diamine substrates, was originally described as a histaminase in 1929 [89]. hDAO is the main enzyme catabolizing ingested histamine [26,90]. Unlike VAP-1, DAO is a soluble protein, which is released from kidney and intestinal epithelial cells and requires an external stimulus [91]. Pathologies that limit DAO secretion from basolateral vesicles, e.g., inflammation and degenerative intestinal disorders, lead to hDAO deficiency [26,92,93]. Furthermore, low DAO levels are known to be caused by genetic mutations. The preferred substrates for DAO are histamine (Km = 2.8 ± 0.07 μM), 1-methylhistamine (Km = 3.4 ± 0.3 μM), and putrescine (Km = 20 ± 1 μM) [25]. Of the DAO substrates, histamine regulates several physiological processes (e.g., gastric acid secretion, central nervous system functioning, hypersensitivity reactions, bronchial asthma, tumorigenesis, and multiple elements of immune regulation). hDAO has a central role in histamine intolerance, which is caused by histamine-rich food, alcohol, or drugs that release histamine or inhibit hDAO and, thus, leads to an excess of histamine and symptoms similar to an allergic reaction in the patients [26]. hDAO deficiency has been described in patients with atopic eczema, chronic urticaria, chronic abdominal pain or inflammatory bowel disease, migraines, and asthma [94,95,96,97,98,99]. Histamine also induces contractions and spontaneous abortions when injected into pregnant animals [28,100,101,102]. The enzymatic activity of DAO increases several 100-fold during pregnancy, and low DAO activity in the first trimester has been associated with a 16-fold increased risk of fetal death [28,103]. Women suffering from early-onset preeclampsia (delivery before week 34 associated with a 10-fold higher risk of mortality [104]) were shown to have significantly lower hDAO levels in the first trimester of pregnancy compared to healthy controls, which is believed to be damaging for the vascular remodeling modulating the uteroplacental blood flow [105]. Based on a mice study, DAO regulates the homeostatic histamine and putrescine levels, which is important in embryo implantation [106]. 

### 5.2. Off-Target Function

Despite strictly controlled development processes, any drug bears a risk of unintended and potentially harmful responses, i.e., adverse drug reactions (ADRs) [107]. Commonly, ADRs are caused by the drug molecule interacting with targets beyond the one(s) it was designed for, so-called off-targets [108]. DAO is an important off-target due to its role as the frontline enzyme for clearing extra-cellular histamine [11,26]. A drug molecule with an off-target activity that causes inhibition of hDAO can cause a rise in histamine levels and related ADRs resembling allergic reactions (e.g., headaches, gastrointestinal disorders, skin reactions, flushing, sneezing, asthma-like wheezing, muscle aches, hypotension, and arrhythmias) and, at its worst, anaphylactic shock. Inhibition of pig and sheep DAO with the potent and irreversible inhibitor aminoguanidine, followed by oral challenge with reasonable levels of histamine, significantly induced anaphylactic shock-like symptoms and increased death rates compared to control animals [27,109]. Relevant to humans, high levels of histamine are found in some common food (e.g., cheese, sausages, yogurts, etc.), in alcohol, and can also be induced by drugs releasing histamine or blocking DAO [26,110]. Inhibition of DAO can also increase the risk of developing intestinal carcinoma, indicating that prolonged use of drugs inhibiting DAO may elevate the cancer risk [111,112]. The patients of atopic eczema, chronic urticaria, chronic abdominal pain or inflammatory bowel disease, migraines, and asthma [94,95,96,97,98,99] that have hDAO deficiency, as well as pregnant women, form high-risk groups for severe ADRs if treated with drugs causing histamine release or drugs having an off-target and activity-blocking effect on hDAO. Currently, aminoguanidine, berenil, pentamidine, and metformin have been indicated to have off-target, activity-blocking effects on hDAO, but many more drugs are expected to show similar behavior, thereby potentially causing severe ADRs [13,27,113].

### 5.3. Unintended Drug–hDAO Interactions at the Atomic Level

Berenil and pentamidine are diamine derivatives, which are drugs for treating trypanosomiasis and pneumocystis pneumonia infections, interact with the minor groove of DNA [114,115]. Both of them have amidophenyl moieties at the ends but vary in the length and chemical nature of the central linkers. The X-ray structures of hDAO in complex with berenil (PDB code 3HIG) and pentamide (PDB code 3HII) revealed the binding mode of the inhibitors [11] (Table 3, Figure 6). Aminoguanidine is a potent inhibitor of DAO with reported IC50 values of ≈ 30 μM [116] and ≈ 150 nM [12]. It is an experimental therapeutic, which prevents advanced glycation end product (AGE) formation [117,118]. Several clinical trials conducted on aminoguanidine indicated that it would be beneficial for treating patients with diabetic nephropathy but also caused many adverse reactions in high doses, e.g., flu-like symptoms, abnormal kidney function tests, and effects on the gastrointestinal tract [117,118,119]. In clinical therapy, the peak plasma concentration of aminoguanidine is ≈ 50 μM [117], which is much higher than the IC50 of hDAO and enough to inhibit the majority of peripheral hVAP-1 (IC50 ≈ 30 μM) as well [12]. Aminoguanidine may, however, be used in small doses for treating diabetic nephropathy as well as other AGE-related disorders 118]. In the X-ray structure complexes, pentamidine and berenil do not interact covalently with the TPQ cofactor (Figure 6A,B) but aminoguanidine forms a covalent adduct with TPQ (Figure 6C). Its guanidinium group makes polar interactions with Asp373 and stacks between Val458, Tyr371, and Trp376 (Figure 6C). Metformin, a drug used for the treatment of type 2 diabetes, causes gastrointestinal side effects for 30%–50% of patients, and 5% has such severe symptoms that they discontinue the usage of metformin [113]. Based on the molecular structure of metformin (Figure 6C), it also interacts covalently with TPQ but likely forms even more interactions.

The binding modes of pentamidine (Figure 6A) and berenil (Figure 6B) resemble each other but also have apparent differences. In these complex structures, TPQ is in on-copper conformation and does not make direct contact with the inhibitors. In both of the inhibitors, the buried amidinium group interacts with the catalytic Asp373, and the connected phenyl ring stacks with Tyr371 and Trp376. Pentamidine binds in a U-shaped form and the other amidinium group forms hydrogen bonds with Ser380 and the main chain of Glu377and Trp376. Its central linker region does not make any significant interactions. Berenil extends straight out from the active site, and the central triazine linker is positioned between Tyr459 and Asp186, and Asp186 makes electrostatic interactions with the nitrogens in the triazine moiety. The second phenyl ring stacks between Phe148 and Phe435 of ArmI. The second amidinium group is exposed to solvent near the surface opening of the active site channel.

## 6. Special Considerations in hVAP-1-Targeted Drug Design

### 6.1. Species-Specific Features in VAP-1 Ligand Recognition 

Due to the fact that rodents are routinely used for estimating inhibitor potency, it is crucial to elucidate the difference in ligand binding among human, mouse, and rat VAP-1. Sequence analysis shows a high degree of similarity between hVAP-1 and rodent VAP-1s: hVAP-1 is 80% identical with rVAP-1 and 83% identical with mVAP-1 [120]. Nevertheless, species differences in ligand binding to VAP-1 were documented using SSAO-activity in plasma, umbilical artery, or homogenized tissues more than two decades ago [121,122]. In contrast to the very first modeling study by Marti et al. [123], who failed to find the sequence basis for the observed species-specific enzymatic properties of VAP-1, Bligt-Lindén et al. [88] showed that hVAP-1 has broader and more hydrophobic active site channel than rodent VAP-1; thus, hVAP-1 should prefer larger and more hydrophobic inhibitors. Indeed, there are no major differences in the active site near TPQ, while the active site channel differs between human and rodent VAP-1 [88]. The different cavity architecture is caused by sequence variations in positions 173, 177, 180, and 761 from one chain and 447 from the other one, which are responsible for the species-specific differences in inhibitor binding (Figure 7A) [88]. Highly species-specific pyridazinone inhibitors **R15**, **R16**, and **R17** with a preference for hVAP-1 were recently structurally studied [10]. Comparison of the X-ray structures of hVAP-1 complexes and modeling of the inhibitor complexes for rodent and monkey VAP-1 suggested that the same residues are responsible for the species-specific inhibitor binding properties of pyridazinones (Figure 7A). Since these residues are scattered all over the active site channel, it is a challenging task to design an equally potent inhibitor towards primate and rodent VAP-1 [10]. 

Modeling study by Inoue et al. [81] suggested that differences in residues 173 (Phe in hVAP-1 vs. Thr in rVAP-1) and 447 (Leu vs. Phe) could be responsible for a species-specific inhibitor potency of **R6** (Figure 7A). Based on their hypothesis, the highly hydrophilic Asp173 in mVAP-1 can be responsible for a high preference for hVAP-1 over mVAP-1 [88]. Foot et al. [54] modeled the binding orientation of compound **R10** that is structurally close to PXS-4728A and revealed the essential role of hydrophobic interactions with Phe173 and Tyr394. Based on their analysis, the species-specificity in position 173 (hydrophobic Phe in hVAP-1, polar Thr in rVAP-1, and negatively charged Asp in mVAP-1; Figure 7A) is a probable reason for a slightly higher potency of PXS-4728A towards hVAP-1 as compared to rodent VAP-1. Kubota et al. [120] recently performed extensive biochemical comparison of inhibitor potency toward human, mouse and rat VAP-1 and revealed apparent species-specific differences: semicarbazide was 10 and 3 times more potent toward hVAP-1 (85.9 µM) as compared to rVAP-1 (993 µM) and mVAP-1 (295 µM), respectively; in contrast, hydralazine more efficiently inhibited rodent VAP-1 than hVAP-1 (1.2 µM for rVAP-1 and 3.1 µM for mVAP-1 vs. 7.8 µM for hVAP-1). Another small inhibitor LJP-1207 showed less species-specificity: it was 2 and 2.5 times more potent towards hVAP-1 than to rVAP-1 and mVAP-1, respectively (252 nM vs. 120 nM and 102 nM). The slightly bulkier inhibitor PXS-4728A showed a weak preference for hVAP-1, while large compound **R7** developed by Astellas showed a slight preference for hVAP-1 (23 nM) over the rat ortholog (50 nM) but much weaker binding to the mouse ortholog (414 nM). As a summary of the previous studies on species-specific binding properties of hVAP-1 inhibitors, it is clear that computational methods should be used to predict the species-specific binding properties in hVAP-1-targeted ligand design. In general, the likelihood of species-specificity increases with enlarged ligand size and is most prominent for those ligands that interact with the active site channel farther away from the catalytic site [1]. 

### 6.2. The Risk for Interaction with Other CAO Sub-Family Members

In addition to species-specific binding properties (Appendix A), differences in the expression profile of the CAO sub-families should be considered when model organisms are selected for in vivo studies [1]. As an example, the pyridazinone inhibitors had a similar in vitro potency against VAP-1 from human and cynomolgus monkey [10], but cynomolgus monkey is not a good model organism for in vivo studies since it also has soluble plasma amine oxidase (SAO; AOC4), the closest CAO family member to VAP-1 [1]. In fact, of all studied primates, only human and gorilla lacked SAO [1]. Of rodents, mice had the same CAO sub-families (VAP-1, DAO, and RAO) as humans, whereas rats have only VAP-1 and DAO [1]. Thus, the best preclinical model organism for hVAP-1-targeted ligand design would be mouse [1] and, naturally, transgenic mouse overexpressing human VAP-1 on endothelium is an even better alternative for in vivo studies to avoid both sub-family cross-reactions and species-specific binding properties [17].

Of the CAO sub-families expressed in human, RAO is also a monoamine oxidase, but it has a different substrate preference profile, and its enzymatic activity has been detected only in the eye [3]. DAO, in turn, is a known off-target and interfering with its normal function as histaminase with drug molecules results in ADRs [11,113,117,118,119]. At the moment, to the best of our knowledge, modeling studies on VAP-1-targeting inhibitors that could result in off-target interactions with DAO have not been published. The X-ray structure of hDAO has been solved in complex with the experimental drug aminoguanidine, which inhibits both DAO and VAP-1. Aminoguanidine inhibits DAO significantly better than VAP-1, but its clinically used concentration also inhibits hVAP-1 [116]. Comparison of the hDAO-aminoguanidine complex with hVAP-1 structure shows that of the interacting residues only Tyr371 is conserved while the Trp376/Phe389, Val458/Leu468, and Tyr459/Leu469 replacements likely contribute to its weaker binding into hVAP-1 (Figure 7B). The position corresponding to Asp186 in hDAO (Figure 7A) is highly conserved within each CAO subfamily (Thr212 in VAP-1, His in RAO and Asn in SAO) and, thus, suggested to be important for the substrate selectivity of CAOs [11]. The corresponding Thr212, together with Thr210 and Tyr394 in hVAP-1 (Phe184 and Val381 in hDAO), is involved in imidazole and pyridazinone binding. In general, the residues in the active site channel of hVAP-1 are not conserved in hDAO, and furthermore, the tip of Arm I has a totally different conformation in these two CAOs. However, it is advisable to test the binding of designed hVAP-1 inhibitors for their effect on hDAO as the catalytic site is highly conserved and small adjustments and rearrangements of side chain conformations in the active site channel might allow their binding to hDAO as well.

## Figures and Tables

**Figure 1 molecules-25-01293-f001:**
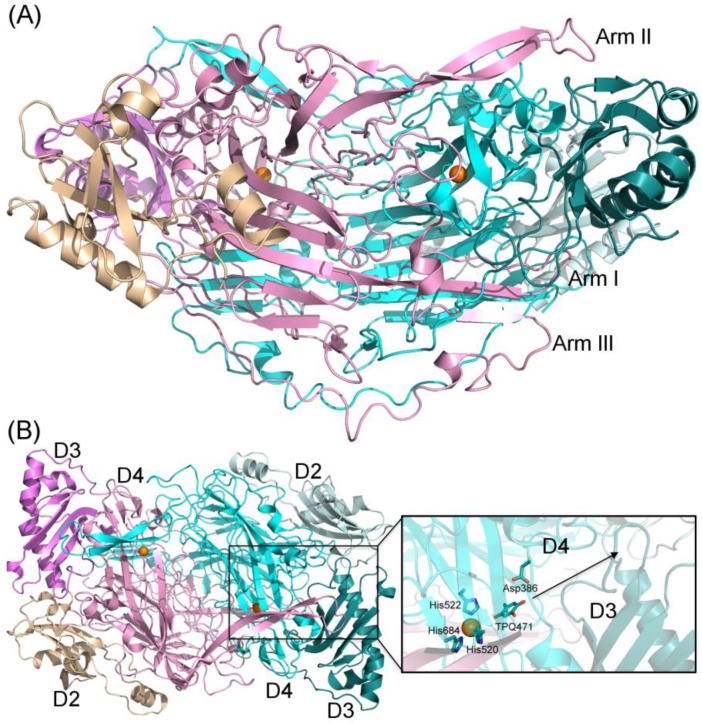
Structural features of human copper-containing amine oxidase (CAO) dimer. (**A**) Side view of the hVAP-1 dimer (PDB code 4BTY). Three long β-hairpin arms stabilize the tight dimer; (**B**) top view of hVAP-1 shows the D2, D3, and D4 domains in each monomer. In the deeply buried active site, the copper ion (orange sphere) is coordinated by the three conserved histidine residues (H520, H522, and H684 in hVAP-1). TPQ (TPQ471) is shown in off-copper, productive conformation, and the catalytic aspartate (Asp386) resides next to the quinone cofactor. Arm I (pink) extending from D4 of monomer B (pink) forms one wall of the active site channel. The black arrow depicts the direction from the catalytic site via the active site channel formed by the D3 and D4 domains to the surface of the protein.

**Figure 2 molecules-25-01293-f002:**
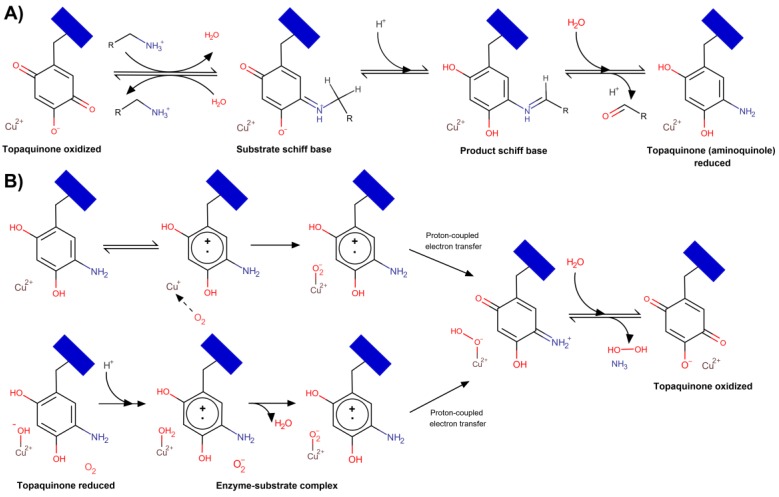
Proposed AOC catalytic mechanism. (**A**) Mechanism of reductive half-reaction; (**B**) two proposed mechanisms of oxidative half-reaction; inner-sphere electron transfer pathway is shown above, while the outer-sphere is below. Protein represented as a blue rectangle. Based on the schemes presented in [31].

**Figure 3 molecules-25-01293-f003:**
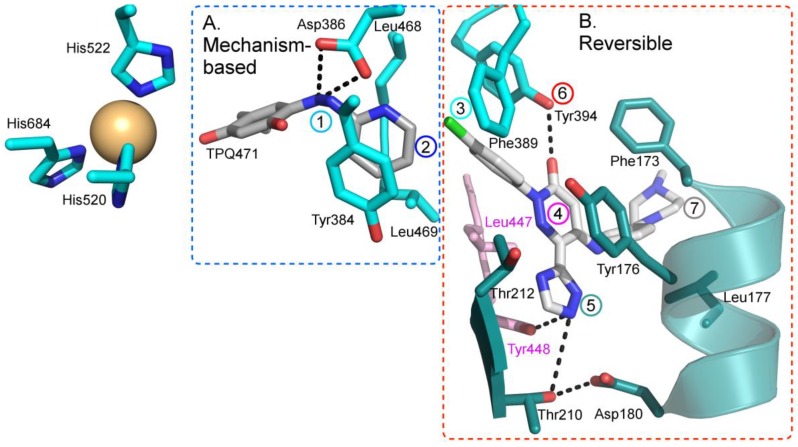
Binding site map for the reviewed hVAP-1 inhibitors. The binding mode of mechanism-based inhibitors is depicted using the 2HP–hVAP-1 complex (PDB code 2C11; (**A**) mechanism-based; boxed in blue) as a representative structure and the binding mode of reversible inhibitors is illustrated using the overlaid pyridazinone–hVAP-1 complex (PDB code 4BTY; (**B**) reversible; boxed in red). Circled numbering 1–7 with color-coding pinpoint the common interaction sites in the catalytic site (blue box) and the active site channel (red box) of hVAP-1. The same coding scheme is used in Table 2.

**Figure 4 molecules-25-01293-f004:**
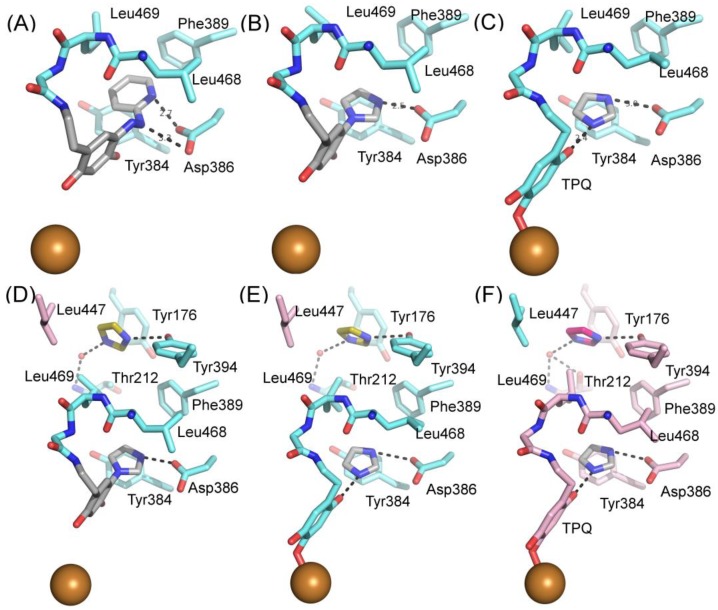
The binding modes of 2HP and imidazoles to hVAP-1. Chain A of the hVAP-1 dimer is shown in pink and chain B in cyan; (**A**) the covalent adduct of 2HP and TPQ (grey; PDB code 2C11); (**B**) the covalent adduct of imidazole and TPQ (grey; PDB code 2Y74); (**C**) the non-covalent complex of imidazole (PDB code 2Y73); (**D**) the binding site of the second imidazole (yellow) in the covalent imidazole complex (PDB code 2Y74); (**E**) the binding site of the second imidazole (yellow) in chain A of the non-covalent imidazole complex (PDB code 2Y73); (**F**) the different binding mode of the second imidazole in chain B of the non-covalent imidazole complex (PDB code 2Y73).

**Figure 5 molecules-25-01293-f005:**
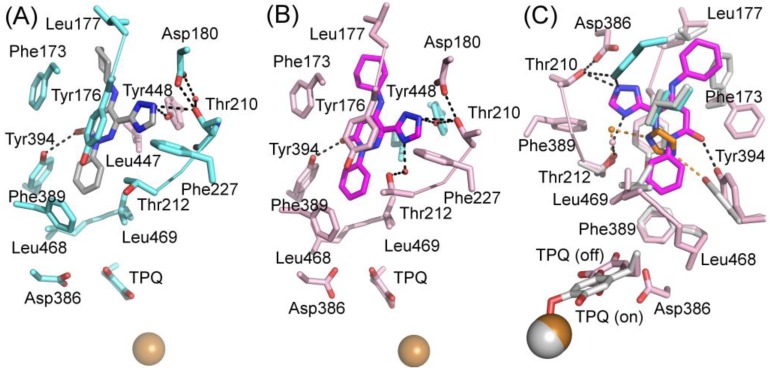
Pyridazinone binding to hVAP-1; (**A**) binding mode of pyridazinone R15 to chain A; (**B**) binding mode of pyridazinone R15 to chain B. Leu447 from the Arm I of chain A (cyan, behind R15) is not labeled; (**C**) comparison of pyridazinone R15 (pink) and the second imidazole (orange) binding to chain B of hVAP-1 (grey). The point of view is rotated to show the hydrogen bonding interactions of R15 (black dashed lines) and imidazole (orange). Labels for Tyr176 (pink), Leu447, and Tyr448 (cyan) stacking with the inhibitors are not shown for clarity.

**Figure 6 molecules-25-01293-f006:**
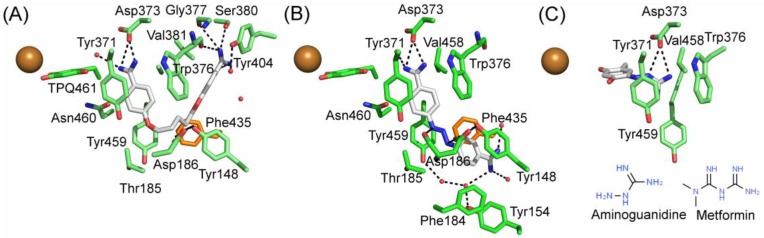
hDAO interactions with off-targets. (**A**) Pentamidine; (**B**) Berenil; (**C**) Aminoguanidine. Phe435 of Arm I from the other monomer is shown in orange.

**Figure 7 molecules-25-01293-f007:**
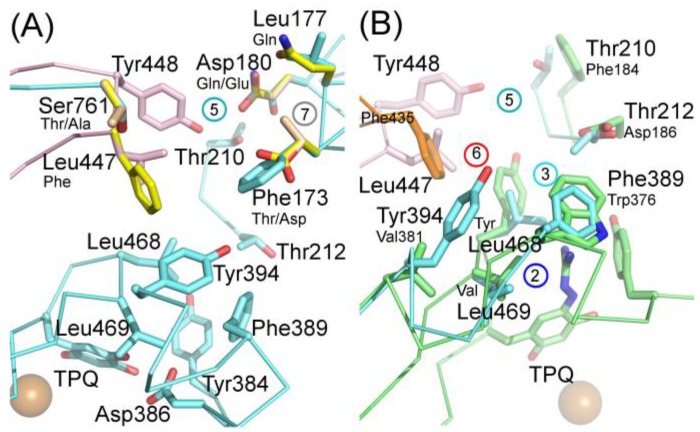
(**A**) Species-specific residues in the VAP-1 active site channel. Residues in rat (wheat)/mouse (yellow) VAP-1 are listed below the residue labels of hVAP-1 (cyan/pink). The catalytic center around TPQ is totally conserved, and the residue differences concentrate on the Arm I and the first helix of D3; (**B**) key residue differences between hVAP-1 and hDAO (green/orange). The Aminoguanidine–hDAO complex (green; PDB code 3MPH) superimposed with hVAP-1 structure (PDB code 2C10). Binding sites depicted similarly as in Figure 3.

**Table 1 molecules-25-01293-t001:** List of hVAP-1 inhibitors in ongoing or completed clinical trials as of 15 January 2020.

Name and Structure	Type of Inhibition	Company	Current Status	Ref.
Hydralazine 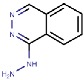	Mechanism-based	Novartis International AG, Switzerland	FDA Approved in 15.01.1953 for the treatment of hypertension; later discontinued due to the development of newer medications.	[47,48]
LJP1207 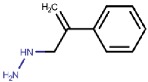	Mechanism-based	La Jolla Pharmaceutical Company, USA	Discontinued; not appropriate for drug development due to its potentially toxic allylhydrazine structure	[49,50,51,52,53]
ASP8232Undisclosed Structure	Unknown	R Tech Ueno Ltd., Japan; Astellas Pharma Europe BV, The Netherlands	Discontinued on Phase2 trials (NCT02302079, NCT02218099) due to strategic prioritization	[44,45]
PXS-4159A (R10) 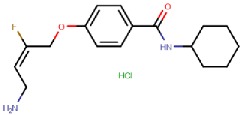	Mechanism-based	Pharmaxis Ltd., Australia	In preclinical toxicology evaluation	[54]
PXS 4728A (BI 1467335) 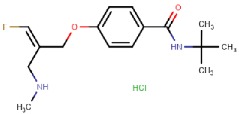	Mechanism-based	Pharmaxis Ltd., Australia	Completed Phase II clinical trials for the treatment of non-alcoholic steatohepatitis (NCT03166735)	[55,56,57]
PXS-4681A 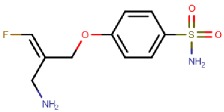	Mechanism-based	Pharmaxis Ltd., Australia	Discontinued	[58,59]
LJP1586 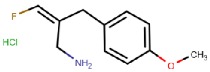	Mechanism-based	La Jolla Pharmaceutical Company, USA	Discontinued	[22,60,61,62,63,64]
TERN-201Undisclosed Structure	Mechanism-based	Eli Lilly and Company; Terns Pharmaceuticals	Phase I clinical trials for the treatment of non-alcoholic steatohepatitis	[65]
PRX167700Undisclosed Structure	Unknown	Cambridge Biotechnology, UK; Proximagen, UK; Roche, Switzerland	Completed Phase II clinical trials for the treatment of osteoarthritis (NCT01945346)	
SzV-1287 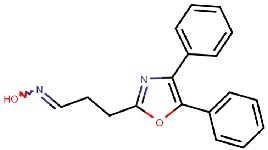	Mechanism-based	Semmelweis University, Hungary	In preclinical studies; patented as a mean to treat hyperalgesia and allodynia in traumatic neuropathy or neurogenic inflammation	[66,67,68,69]
U-V002 (R7) 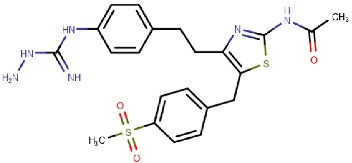	Mechanism-based	R Tech Ueno Ltd., Japan;	Unknown	[70,71,72,73]
BTT-2027 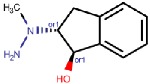	Mechanism-based	Biotie Therapies Corp.	Discontinued	[20]
BTT2052 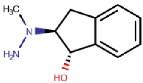	Mechanism-based	Biotie Therapies Corp.	Discontinued	[74]

**Table 2 molecules-25-01293-t002:** List of computational studies modeling the interactions between various inhibitors and hVAP-1.

Inhibitor	Type of Inhibitor, IC50	Used VAP-1 Structure	Docking Tool	Ref.
**Hydrazine Alcohols**				
**R1**: Compound **2a**,**b** 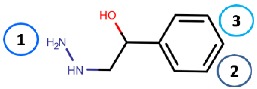	Mechanism-based, 0.04–0.15 µM	2C11	GOLD 3.1.1	[78]
*Interacting residues*: Covalent bond between hydrazine nitrogen and TPQ; H-bonds between hydrazine nitrogen and the hydroxyl group of R1 with TPQ and D386; hydrophobic interactions with Y384, F389, Y394, L468, L469.				
**R2**: Compound **8** 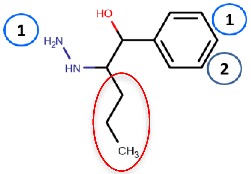	Mechanism-based, 1.2 µM	2C11	GOLD 3.1.1	[78]
*Interacting residues*: Covalent bond between hydrazine nitrogen and TPQ; H-bonds between nitrogen and TPQ and D386, and between the hydroxyl group and D386; hydrophobic interactions with Y384, Y394, F389.				
**R3**: Compound **12** 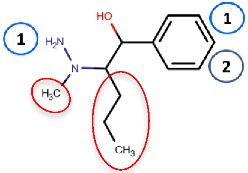	Mechanism-based, 0.26 µM	2C11	GOLD 3.1.1	[78]
*Interacting residues*: Covalent bond between hydrazine nitrogen and TPQ; H-bonds between the hydroxyl group and D386; hydrophobic interactions with Y384, Y394, F389.				
**R4**: Compound **11a**–**d** 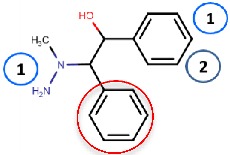	Mechanism-based, 0.28–1.18 µM	2C11	GOLD 3.1.1	[78]
*Interacting residues*: Covalent bond between hydrazine nitrogen and TPQ; 11a,b form H-bond with TPQ, while 11c,d with D386; hydrophobic interactions with A370, Y384, F389, Y394, L468, L469; 11c,d form π–π interaction with Y372 or Y384.				
**Thiazoles**				
**R5**: Compound **10** 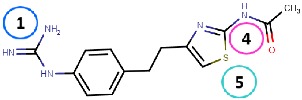	Mechanism-based, 0.23 µM	2C11	GOLD 5.0	[79]
*Interacting residues*: Covalent bond between guanidine group and TPQ; H-bond between guanidine group and D386; S–O interaction of sulfur atom in the thiazole ring with T210 backbone carbonyl oxygen; proton–π interaction of amide moiety with Y176.				
**R6**: Compound **35a** 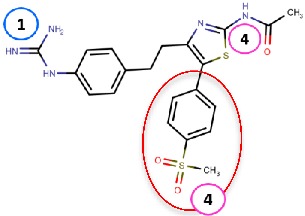	Mechanism-based, 34 nM	2C11	GOLD 5.0	[80]
*Interacting residues*: Covalent bond between guanidine group and TPQ; π–proton interactions of sulfonyl moiety with L447, amide nitrogen with Y176, and guanidine nitrogen with Y384.				
**R7**: Compound **35c** 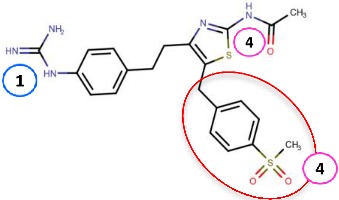	Mechanism-based, 20 nM	2C11	GOLD 5.0	[80]
*Interacting residues*: Covalent bond between guanidine group and TPQ; H-bonds between sulfonyl moiety with L447 and D446, π–proton interactions of sulfonyl moiety with L447, amide nitrogen with Y176, and guanidine nitrogen with Y384.				
**Indanols**				
BTT2052 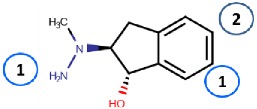	Mechanism-based, 0.06 µM	1US1	Manual docking	[74]
*Interacting residues*: Covalent bond between hydrazine nitrogen and TPQ; H-bond between the hydroxyl group and D386; hydrophobic interactions with M211, Y384, F389, L469.				
**1*H*-imidazol-2-amines**				
**R8**: Compound **19** 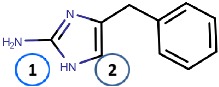	Mechanism-based, 32 µM	2C11	GOLD 5.1	[81]
*Interacting residues*: Covalent bond between the amine group and TPQ; H-bond between NH of the imidazole ring with N470; π–π interaction between imidazole ring and Y384; π–proton interaction of benzene ring with L469.				
**R9**: Compound **37b** 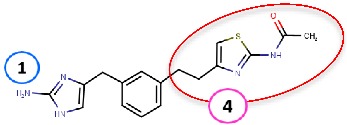	Mechanism-based, 19 nM	2C11	GOLD 5.1	[81]
*Interacting residues*: Covalent bond between the amine group and TPQ; π–proton interactions of thiazole group with T212 and L447; S-O interaction of thiazole sulfur with T210 backbone carbonyl oxygen; proton–π interaction between amide moiety and Y176; CH–O interaction of acetyl moiety and D180.				
**Allylamines**				
**R10**: Compound **28** 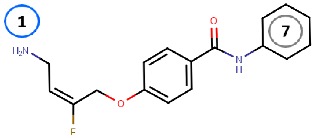	Mechanism-based, 10 nM	?	MOE 2011.10	[54]
*Interacting residues*: Covalent bond between the amine group and TPQ; Hydrophobic interactions with F173 and Y394.				
**Phenyl-piperidinyl amine**				
**R11**: ELP12 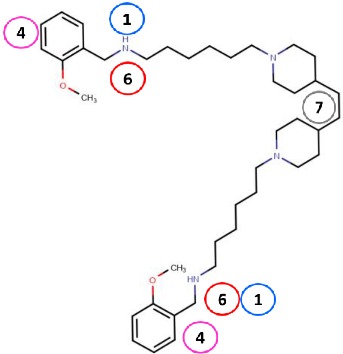	Reversible, Ki_(EI)_111 µM/Ki_(ESI)_ = 163 µM	2C10	AutoDock 4.2	[82]
*Interacting residues*: Salt bridges between amino groups of **R11** with the carboxylates of D180 and D446 and H-bonds with the side-chain of Y448 and with the backbone carbonyl of T424; an aromatic ring is stacked to the ring of Y176; hydrophobic interactions with F173, Y394, P397, I425, and L447; H-bond with Y394.				
**Glycine amides**				
**R12**: Compound **4a** 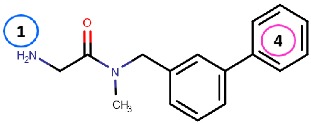	Mechanism-based, 0.80 µM	2C11	GOLD 5.2	[83]
*Interacting residues*: Covalent bond between the primary amine group and TPQ; CH–p interaction with L447, CH–O interaction with the L468 backbone carbonyl oxygen, arene--H interaction with L447.				
**R13**: Compound **4g** 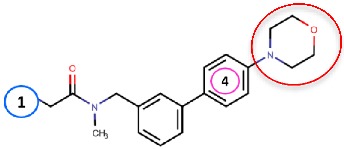	Mechanism-based, 0.34 µM	2C11	GOLD 5.2	[83]
*Interacting residues*: Covalent bond between the primary amine group and TPQ; CH–O interaction with the L468 backbone carbonyl oxygen, arene–H interaction with L447.				
**R14**: Compound **17h** 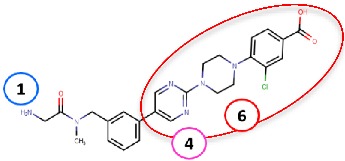	Mechanism-based, 25 nM	2C11	GOLD 5.2	[84]
*Interacting residues*: Covalent bond between primary amine group and TPQ; Three CH–p interactions: central phenyl ring with L469, pyrimidine ring with L447, and the hydrogen in the piperazine with Y394; two CH–O interactions: the oxygen in the glycine amide moiety with L468, and the hydrogen of the carbonyl group in the glycine amide moiety with the carbonyl oxygen of the L468 backbone; chloro group formed a halogen–O interaction with the carbonyl oxygen of the D446 backbone.				

**Table 3 molecules-25-01293-t003:** A comprehensive list of experimentally solved structures of hVAP-1 and diamine oxidase (DAO) deposited into the Protein Data Bank as of 15 January 2020.

PDB ID	Resolution,Å	Expression System	Date Deposited	Ligands	Inhibition Type,IC50/Ki	Ref.
	**Vascular Adhesion Protein 1**
1PU4	3.20	*Cricetulus griseus*(CHO cells)	24.06.03	—	n.a.^1^	[8]
1US1	2.90	*Cricetulus griseus*(CHO cells)	17.11.03	—	n.a.	[8]
2C10	2.50	*Homo sapiens* (HEK293 cells)	09.09.05	—	n.a.	[7]
2C11	2.90	*H. sapiens* (HEK293 cells)	09.09.05	2-Hydrazinopyridine 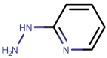	Mechanism- based, n.d.^2^	[7]
2Y73	2.60	*H. sapiens* (human serum)	28.01.11	Imidazole 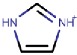	Mechanism- based, n.d.^2^	[9]
2Y74	2.95	*H. sapiens* (human serum)	28.01.11	Imidazole 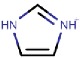	Mechanism- based, n.d.^2^	[9]
3ALA	2.90	*Cricetulus griseus*(CHO cells)	29.07.10	—	n.a.	[85]
4BTW	2.80	*H. sapiens* (human serum)	19.06.13	**R15**: 5-(cyclohexylamino)-2-phenyl-6-(1*H*-1,2,4-triazol-5-yl)-3(2*H*)-pyridazinone 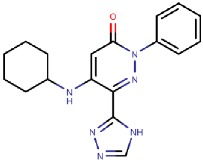	ReversibleIC50 = 71 nM	[10]
4BTX	2.78	*H. sapiens* (human serum)	19.06.13	**R16**: 5-isopropylamino-2-phenyl-6-(1*H*-1,2,4-triazol-5-yl)-3(2*H*)-pyridazinone 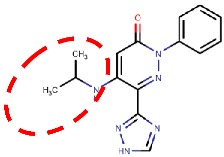	ReversibleIC50 = 290 nM	[10]
4BTY	3.10	*H. sapiens* (human serum)	19.06.13	**R17**: 5-[4-(4-methylpiperazin-1-yl)phenylamino]-2-(4-chlorophenyl)-6-(1*H*-1,2,4-triazol-5-yl)-3(2*H*)-pyridazinone 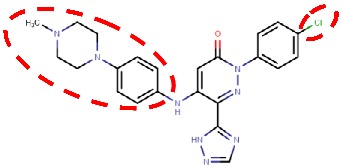	ReversibleIC50 = 20 nM	[10]
	**Diamine Oxidase**
3HI7	1.80	*Drosophila melanogaster* (S2 cells)	19.05.09	—	n.a.	[11]
3HIG	2.09	*Drosophila melanogaster* (S2 cells)	19.05.09	Berenil (4-[(2*E*)-3-(4-carbamimidoylphenyl)triaz-2-en-1-yl]benzene-1-carboximidamide) 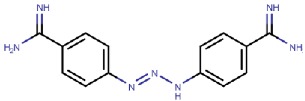	ReversibleKi = 13 nM	[11]
3HII	2.15	*Drosophila melanogaster* (S2 cells)	20.05.09	Pentamidine (1,5-*bis*(4-amidinophenoxy)pentane) 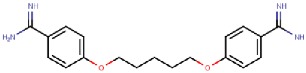	ReversibleKi = 290 nM	[11]
3K5T	2.11	*Drosophila melanogaster* (S2 cells)	08.10.09	—	n.a.	[13]
3MPH	2.05	*Drosophila melanogaster* (S2 cells)	27.04.10	Aminoguanidine 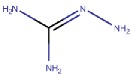	Mechanism- based,Ki = 140 nM	[12]

^1^ Not applicable due to the absence of inhibitors in the crystallographic unit. ^2^ Inhibition constants are not known.

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
