# Peer review of "Human Copper-Containing Amine Oxidases in Drug Design and Development"

_molecules, 2020, doi:10.3390/molecules25061293_

Round 1
Reviewer 1 Report
Albeit I'm not an expert of amine oxidases, this review is rather useful, in my opinion. Yet, it'll need some restructuring - in particular, please introduce a numbering of all the structures you show (perhaps prefix it with R like "Revue", R1, R2, ..., in order to avoid confusion with compounds you refer at by the numbers given in the original papers). Then, you could provide a table (Ri, IUPAC name) as supplementary information, if really needed (not in my opinion - furthermore, you're not showing any new compounds, so let their "parents" give their IUPAC names). Another issue is the strange concept of "mechanism-based" ?? inhibition. I'm afraid ALL acts of inhibition are based on... some kind of mechanism, right? It may imply "negociating" the way into the active site, formation of key interactions (covalent or not), inducing conformational changes or not. I suspect you call "mechanism-based" inhibitors what people usually refer to as "suicidal" inhibitors, e.g. irreversible covalent binders! If you dislike the term "suicidal", just stick to "irreversible covalent". On the other hand... how can you measure an IC50 of an irreversible covalent inhibitor? Makes little sense - the only meaningful term would be the Kon rate! Reversible covalent inhibitors - ok, those can reach an equibibrium on/off. Please be specific on these issues.
English is globally fine - yet often convoluted and weird: most of al in the Abstract. Let us have a look the first phrase: "In humans, two members of the copper-containing amine oxidase family are of importance for human health and disease". Ah! In humans, enzymes are important for HUMAN health - not rat health? Who would have guesed that? And - very specifically: they are important for health AND DISEASE, as if the two were not antonyms. But - best of the best - TWO enzymes are important: all the others were just added to the human body by a sloppy Darwininan evolution process, to idle around all day long !? Should I believe this? Moreover, a "3D dimensional space" is a nonsense, as D already stands for "dimensional". Otherwise, you'd also say "CIA Agency or UNO Organization". All in all, some proofreading would greatly improve the manuscript!
Reviewer 2 Report
This really well conceived and deeply discussed. The authors could only mention some information about the role of the in silico prediction of admet properties.
Differences of key contacts within protein isoforms or ortologues could be summarized as tables.
Author Response
Point1: This really well conceived and deeply discussed. The authors could
only mention some information about the role of the in silico prediction of
admet properties.
Response1: We have not found any publications on copper-‐‑containing amine oxidases where the in silico prediction of the ADMET properties of the inhibitors would have reported. Thus, we could not write about the role of the predictions.
Point2: Differences of key contacts within protein isoforms or ortologues could
be summarized as tables.
Response2: As suggested by the Reviewer, we have now prepared a table
listing the variable residues in the VAP-‐‑1 orthologs and/or in the copper-‐containing amine oxidase sub-‐‑families that are involved in inhibitor binding (Lines 687-‐‑688: Table S2: List of amino acids relevant to inhibitor binding that differ among VAP-‐‑1 orthologs and copper-‐‑containing amine oxidase sub-families.)